# Anti-Virulence Properties of Plant Species: Correlation between In Vitro Activity and Efficacy in a Murine Model of Bacterial Infection

**DOI:** 10.3390/microorganisms9122424

**Published:** 2021-11-25

**Authors:** José Luis Díaz-Núñez, Macrina Pérez-López, Norma Espinosa, Nayelli Campos-Hernández, Rodolfo García-Contreras, Miguel Díaz-Guerrero, Humberto Cortes-López, Monserrat Vázquez-Sánchez, Héctor Quezada, Mariano Martínez-Vázquez, Ramón Marcos Soto-Hernández, Mireya Burgos-Hernández, Bertha González-Pedrajo, Israel Castillo-Juárez

**Affiliations:** 1Laboratorio de Fitoquímica, Posgrado de Botánica, Colegio de Postgraduados, Texcoco 56230, Mexico; alucard_d_n@hotmail.com (J.L.D.-N.); maky1117@hotmail.com (M.P.-L.); camposh.nayelli@gmail.com (N.C.-H.); b_et_ocl@hotmail.com (H.C.-L.); vazquez.monserrat@colpos.mx (M.V.-S.); msoto@colpos.mx (R.M.S.-H.); burgos.mireya@colpos.mx (M.B.-H.); 2Departamento de Genética Molecular, Instituto de Fisiología Celular, Universidad Nacional Autónoma de México, Ciudad de México 04510, Mexico; nespino@ifc.unam.mx (N.E.); madiaz@ifc.unam.mx (M.D.-G.); 3Departamento de Microbiología y Parasitología, Facultad de Medicina, Universidad Nacional Autónoma de México, Ciudad de México 04510, Mexico; rgarc@bq.unam.mx; 4Laboratorio de Investigación en Inmunología y Proteómica, Hospital Infantil de México Federico Gómez, Ciudad de México 06720, Mexico; hquezadap@yahoo.com.mx; 5Departamento de Productos Naturales, Instituto de Química, Universidad Nacional Autόnoma de México, Ciudad de México 04510, Mexico; marvaz@unam.mx

**Keywords:** cutaneous infection model, type III secretion system, quorum sensing, *Chromobacterium violaceum*, *Pseudomonas aeruginosa*, *Diphysa americana*, *Hibiscus sabdariffa*

## Abstract

Several plant extracts exhibit anti-virulence properties due to the interruption of bacterial quorum sensing (QS). However, studies on their effects at the preclinical level are scarce. Here, we used a murine model of abscess/necrosis induced by *Pseudomonas aeruginosa* to evaluate the anti-pathogenic efficacy of 24 plant extracts at a sub-inhibitory concentration. We analyzed their ability to inhibit QS-regulated virulence factors such as swarming, pyocyanin production, and secretion of the ExoU toxin via the type III secretion system (T3SS). Five of the seven extracts with the best anti-pathogenic activity reduced ExoU secretion, and the extracts of *Diphysa americana* and *Hibiscus sabdariffa* were identified as the most active. Therefore, the abscess/necrosis model allows identification of plant extracts that have the capacity to reduce pathogenicity of *P. aeruginosa*. Furthermore, we evaluated the activity of the plant extracts on *Chromobacterium violaceum*. T3SS (Δ*escU*) and QS (Δ*cviI*) mutant strains were assessed in both the abscess/necrosis and sepsis models. Only the Δ*escU* strain had lower pathogenicity in the animal models, although no activity of plant extracts was observed. These results demonstrate differences between the anti-virulence activity recorded in vitro and pathogenicity in vivo and between the roles of QS and T3S systems as virulence determinants.

## 1. Introduction

Bacteria are social microorganisms that use quorum sensing (QS) to communicate and induce multicellular behaviors [1]. QS is dependent on cell density and is a mechanism by which bacteria release chemical signals (called autoinducers) to their microenvironment to perceive the presence of other cells [2]. This phenomenon allows bacteria to regulate expression of genes that control production of various metabolites and virulence factors, which is the reason it is considered an important target to block bacterial pathogenicity [3]. Thus far, plant species have positioned themselves as one of the primary sources of anti-virulence substances, but one of the most significant challenges is to confirm their anti-pathogenic efficacy at the preclinical and clinical level [4,5,6,7].

Although there are reports of phytochemicals with anti-virulence properties that reduce the pathogenicity of *Pseudomonas aeruginosa* in some animal models (*Caenorhabditis elegans*, *Danio rerio*, and *Galleria mellonella*), in the case of murine models, the reports are scarce [3,8]. In *P. aeruginosa*, a reduction in pathogenicity due to QS inhibition has been reported in some mouse models, such as induced thermal injury, lung infection, and foreign-body infection [9,10,11,12]. Mainly, the reduction of pathogenicity of QS mutant strains relative to their parental strains [9,10,11] and the decreased virulence promoted by some molecules such as furanones [13,14] have been demonstrated. In the case of phytochemicals, the prophylactic administration of synthetic ajoene [(*E*,*Z*)-4,5,9-trithiadodeca-1,6,11-triene 9 oxide] in the PAO1 lung infection model reduced bacterial load by 500 times in the lungs relative to untreated mice [14], while in the urinary tract infection model, oral administration of garlic significantly reduced bacterial load and prevented kidney damage [15].

Moreover, several screens have also been carried out using biosensor strains that contain a QS genetic circuit and an indicator gene, which produces pigments or fluorescence to facilitate quantification of QS inhibition [16]. In most studies evaluating anti-virulence activity with substances of plant origin, inhibition of pigment biosynthesis regulated by QS, such as violacein in *Chromobacterium violaceum* and pyocyanin in *P. aeruginosa*, has been commonly used [17,18,19]. To a lesser extent, other relevant targets such as type III secretion systems (T3SSs) [20,21], key enzymes [22], toxins [23], or two-component systems, among others, have also been evaluated [24,25,26,27,28].

*P. aeruginosa* is an opportunistic pathogen that causes nosocomial outbreaks that are difficult to control due to the presence of antibiotic-resistant strains [29,30]. Various regulatory mechanisms participate in its pathogenesis, e.g., three QS systems called LasI/LasR, RhlI/RhlR, and PQS [31]. QS controls the acquisition of iron through pyochelin and pyoverdine, the release of toxins such as phenazines and hydrogen cyanide, the production of pyocyanin, alginate, and lipopolysaccharides, as well as swarming and biofilm formation, among others [32]. Autoinducers of the diffusible signal factor (DSF) family also participate in this process through the PA1396 sensor kinase, which perceives DSFs released by other bacteria into the medium [33,34,35].

In addition, *P. aeruginosa* produces a DSF called *cis*-2-decenoic acid (CDA) that regulates swarming and pyoverdine production, induces biofilm dispersion, and promotes systemic infection in mice [36,37]. This bacterium also has a T3SS that translocates the effector proteins ExoY, ExoT, ExoS, and ExoU into host cells [38,39]. ExoU is a phospholipase A2 that causes host cell lysis and is the most toxic effector, thus it is recognized as a critical anti-virulence target [40].

*C. violaceum* is a saprophytic soil bacterium and an occasional opportunistic pathogen in humans [41]. It has a QS system called CviI/CviR, which consists of the CviI synthase and the cytoplasmic transcriptional regulator CviR [42]. This system regulates the synthesis of the pigment violacein and some virulence factors such as production of protease, chitinase, formation of biofilm, swarming [43], and the type VI secretion system [42,44,45]. Similarly, it has a T3SS located on the *Cpi-1/-1a* pathogenicity island, which is necessary for bacterial pathogenicity in a sepsis model in mice [46].

Among the challenges facing the development of anti-virulence therapies is verification of the beneficial effect at the preclinical and clinical levels [3,7]. Therefore, the purpose of our study was to evaluate the anti-pathogenic properties of 24 plant extracts at sub-inhibitory concentrations in murine infection models, and to analyze the participation of QS and T3SS. Although there is no consensus on the most suitable murine infection model to evaluate anti-virulence substances [8], it has been proposed that the abscess/necrosis model is reproducible with a small number of animals and allows infection by different bacterial species, including *P. aeruginosa* [47,48]. Using similar models, the identification of some anti-virulence triterpenes against *Staphylococcus aureus* has been reported [49,50,51]. Likewise, we have recently identified hibiscus acid isolated from *Hibiscus sabdariffa* as a compound with anti-pathogenic activity using the abscess/necrosis model [19].

In this work, we identify plant extracts that reduce the pathogenicity of *P. aeruginosa* at sub-inhibitory concentrations and others that stimulate it. However, of the two phenotypes evaluated in vitro (swarming and pyocyanin), only swarming inhibition correlated with anti-pathogenic activity in vivo. In addition, most of the extracts with better activity also reduced ExoU secretion.

Moreover, to the best of our knowledge, this is the first report to analyze the pathogenicity of a Δ*cviI* (CVO26) mutant in murine models. We showed that the CviI synthase is not a pathogenicity determinant compared to the T3SS of this bacterium. Our results reveal substantial differences that must be considered when using these two bacterial species to identify anti-virulence substances and their usefulness for preclinical analysis.

## 2. Materials and Methods

### 2.1. Plant Materials and Extract Preparation

The plants were collected in Mexico, and the specimens were deposited in the Herbarium-Hortorio CHAPA of the Colegio de Postgraduados (Appendix A). The plant material was dried at room temperature for one week, ground, and degreased with hexane (1:10, *w*/*v*) (J.T. Baker^®^, Belmont, NC, USA). After removing the solvent under reduced pressure (Buchi-R114, Flawil, Switzerland), the residues were macerated three times every 24 h with dichloromethane (1:10, *w*/*v*) (J.T. Baker^®^, Belmont, NC, USA). Finally, the supernatant was filtered (filter paper, Whatman International Ltd., Maidstone, England) and the solvent was removed under reduced pressure (Buchi-R114, Flawil, Switzerland) and stored in dark flasks at 4 °C until use. For the evaluation of biological activity, the samples were dissolved in dimethyl sulfoxide (DMSO) (Sigma-Aldrich, St. Louis, MO, USA), and by growth curves or by plate count, it was determined that the concentrations used did not affect bacterial growth.

#### Quantitative Determination of Flavonoids and Triterpenoids

The extracts were diluted in methanol (HPLC, J.T. Baker^®^, Belmont, NC, USA) at 0.25 mg/mL. For quantification of flavonoids, the aluminum chloride colorimetric method (ACM) was used [52], while triterpenoids were determined by the reaction of vanillin and with perchloric acid [53] (Appendix A). In the ACM, 0.5 mL of extract was added to the reaction mixture that contained 2.8 mL of distilled water, 1.5 mL of 95% ethanol, 0.1 mL of 1 M potassium acetate, and 0.1 mL of 10% aluminum chloride. It was incubated at room temperature for 30 min, and the absorbance at 415 nm was measured. To quantify the triterpenoids, 0.03 mL of the extract was dissolved in 0.1 mL of ethanol and concentrated in a water bath (90 °C). Subsequently, 0.4 mL of glacial acetic acid and 0.1 mL 5% vanillin were added. The solution was heated for 30 min at 60 °C and cooled in an ice-water bath. Finally, 3 mL of glacial acetic acid was added, and the absorbance at 550 nm was measured. A calibration curve was used to calculate the concentrations of flavonoids and triterpenoids, and they were expressed in milligrams of quercetin or ursolic acid/g of extract (Appendix A).

HPLC analysis of the *D. americana* extract was carried out on an Agilent 1260 Infinity equipped with an autosampler and a diode array detector (Agilent Technologies, Santa Clara, CA, USA). An ODS C18 column (125 × 4.6 mm, 5 μm, Thermo Scientific^®^, Hypersil^®^, Walthan, MA, USA) was used, and as mobile phase, an acetonitrile/water gradient (pH 2.5) with a flow rate of 1 mL/min at 30 °C was used with a detection wavelength of 330 nm.

### 2.2. Bacterial Strains and Culture Conditions

The *P. aeruginosa* PA14 wild-type (WT) strain and derived mutants (Δ*lasR*/Δ*rhlR* and Δ*pscC*) used in this study are listed in Appendix A. In the case of *C. violaceum*, the ATCC^®^ 31532™ (The American Type Culture Collection, Manassas, VA, USA) WT and the mutant Δ*cviI* (CV026, CDBB1423) strains were obtained from Colección Nacional de Cepas Microbianas y Cultivos Celulares, CINVESTAV (CINVESTAV-IPN, CDMX, Mexico) (Appendix A).

#### Construction of the *C. violaceum* Mutant Strain Δ*escU*

The Δ*escU* mutant strain was constructed by allelic exchange using the recombinant suicide plasmid pRE112. Briefly, the pRE112 plasmid containing 770 bp of DNA upstream and 896 bp of DNA downstream of the *escU* gene was conjugated from the *E. coli* SM10 λ pir (ATCC^®^ 87450TM, The American Type Culture Collection, Manassas, VA, USA) donor strain into the *C. violaceum* WT strain. Colonies from the first homologous recombination event were selected for chloramphenicol resistance (15 μg/mL) and for violacein production (purple colonies). The second event of recombination was selected on LB plates containing 5% sucrose. Elimination of the *escU* gene from the chromosome was corroborated by PCR.

### 2.3. Swarming Inhibition

Swarming inhibition was determined as previously reported [54]. Three milliliters of M9 (0.6% agar) was added per well with the different treatments in 6-well plates (Corning^®^, Corning, NY, USA) and allowed to dry under sterile conditions for 45 min. Subsequently, 25 μL of an overnight culture of *P. aeruginosa* was adjusted to an O.D._600 nm_ = 0.08 and inoculated in the center of each well. The plates were incubated at 37 °C for 24 h in a humid chamber (Thermo Scientific^®^, Walthan, MA, USA), and the ImageJ^®^ program (Bethesda, MD, USA) was used to calculate the swarming area.

### 2.4. Pyocyanin Production

Pyocyanin was determined as previously reported [55]. Bacterial cultures (O.D._600 nm_ = 0.08) were centrifuged at 4000× *g* for 3 min, and 0.8 mL of supernatants was collected. The supernatants were vortexed with 0.42 mL of chloroform (J.T. Baker^®^, Belmont, NC, USA) for 2 min. The samples were centrifuged at 4000× *g* for 8 min. The organic phase was collected and added to 0.8 mL of 0.2 N HCl. The samples were vigorously shaken for 1 min (vortex 2, IKA^®^, Walthan, MA, USA). Then, 0.65 mL of the aqueous phase was taken, and 0.65 mL distilled water was added. Pyocyanin was determined at 520 nm. The data were normalized to cell density.

### 2.5. Type III Secretion Assay

Overnight cultures of *P. aeruginosa* grown at 37 °C and 250 rpm in LB medium were used to inoculate (1:200) modified LB medium supplemented with 200 mM NaCl, 10 mM MgCl_2_, 0.5 mM CaCl_2_, and 5 mM EGTA. The bacterial cultures were grown under the same conditions in the absence or presence of 200 μg/mL of the dichloromethane extracts, until reaching an O.D._600 nm_ = 0.8. Phenoxyacetamide 25 µM (MBX 1641) (ChemBridge, San Diego, CA, USA) was used as a positive control [48]. Each culture was centrifuged at 18,100× *g* for 2 min at 4 °C. The supernatants were centrifuged once again and transferred to clean tubes. Secreted proteins were precipitated from the supernatant using 10% trichloroacetic acid (TCA) and centrifuged at 18,100× *g* for 30 min at 4 °C. The resulting bacterial and protein pellets were resuspended in Laemmli SDS loading buffer, which for the supernatant samples contained 10% saturated Tris to neutralize residual TCA. All samples were normalized according to each culture O.D. and separated on 15% polyacrylamide gels under denaturing conditions. Proteins were detected by Western blot analysis using anti-ExoU polyclonal antibodies and a chemiluminescent detection system (Merck-Millipore, Darmstadt, Germany). Western blot images were quantified densitometrically using the Image Studio Lite software (LI-COR, Lincoln, NE, USA).

### 2.6. Virulence Factors Regulated by QS in C. violaceum

Overnight cultures of *C. violaceum* 31532 WT and Δ*escU* and Δ*cviI* mutant strains were adjusted to an O.D._660 nm_ = 0.1 (Spectronic^®^ Genesys™ 5, Texas city, TX, USA) and incubated in multiwell plates (Corning^®^, Corning, NY, USA) for 18 h. The supernatants were obtained by centrifugation at 4000× *g* for 3 min and were used to determine proteolytic, hemolytic, and chitinolytic activity. The plates were incubated for 48 h at 28 °C, and a digital caliper (STEREN^®^, CDMX, Mexico) was used to measure the areas of activity.

#### 2.6.1. Proteolytic, Hemolytic, and Chitinolytic Activity

The evaluation was carried out as previously reported. For the case of hemolytic and protease activity, ten microliters of supernatant was added to the wells previously prepared on 5% sheep blood agar plates and milk agar plates, respectively [56]. For the case of chitinase activity, it was added to wells in M9 medium plates (Difco Laboratories, Detroit, MI, USA) with 2% colloidal chitin (Sigma-Aldrich, St. Louis, MO, USA) [57].

#### 2.6.2. Biofilm Formation

Overnight cultures were adjusted to an O.D._660 nm_ = 0.1, transferred to glass tubes, and incubated for 48 h at 28 °C [43]. Subsequently, the tubes were washed with distilled water and dried at 40 °C. After adding 1 mL of 0.1% crystal violet and incubating for 20 min, this same procedure was carried out. Finally, the crystal violet was solubilized with 1 mL of 80% ethanol, and the absorbance at 570 nm was measured. The data were normalized with bacterial growth at 660 nm.

#### 2.6.3. Swarming Motility

Overnight cultures were adjusted to an O.D._660 nm_ = 0.05, and 2 μL of inoculum was placed in the center of the semisolid plate (0.25% agar) [58], which was incubated for 24 h in a moist chamber at 28 °C (Thermo Scientific^®^, Walthan, MA, USA), and the displacement area was measured.

### 2.7. Animal Studies

Six- to eight-week-old CD-1 mice (equal ratio of males and females) were obtained from the Facultad de Estudios Superiores, Cuautitlán-UNAM. The animals were kept under standard conditions (23 °C ± 2 °C) with a 12 h light–dark cycle and free access to food (Lab rodent diet 5001, LabDiet^®^, Saint Louis, MO, USA) and water.

#### 2.7.1. Abscess/Necrosis Model with *P. aeruginosa*

Overnight cultures (37 °C) were adjusted to an O.D._600 nm_ = 0.06, incubated until reaching an O.D._600 nm_ = 1–1.5 and adjusted to a final absorbance of 0.08 (Spectronic^®^ Genesys™ 5, Texas city, TX, USA). The cultures were centrifuged at 4000× *g* for 5 min, and the bacterial pellets were washed with PBS (three times) to obtain 60 μL of inoculum containing 10^7^ CFU of *P. aeruginosa*. The inoculum was mixed with the plant extracts, and bacterial viability was determined by plate count. Additionally, to determine the toxicity of some extracts, they were injected without bacteria into the animals (Appendix A). Inoculation was carried out following a procedure reported elsewhere [47,48]. Mice were anesthetized with 64 mg/kg sodium pentobarbital (Pisabental, PiSa^®^ Agropecuaria, Hidalgo, Mexico) and the right thigh depilated with an electric razor and chemical depilator (Remove-Loquay^®^, CDMX, Mexico). One day later, the animals were re-anesthetized and injected subcutaneously with 60 µL of the inoculum.

The abscess area was measured at 24 h and the area of necrosis at 48 h with a digital caliper (STEREN^®^, CDMX, Mexico). On the fourth day, the animals were anesthetized and sacrificed by cervical dislocation. Lesions and livers were removed to calculate CFU/g of tissue by plate count.

#### 2.7.2. Sepsis and Abscess/Necrosis Model with *C. violaceum*

The bacterial inoculum was prepared from overnight cultures incubated at 28 °C and adjusted to an O.D._600 nm_ = 0.5 (Spectronic^®^ Genesys™ 5, Texas City, TX, USA). The cultures were centrifuged at 4000× *g* for 5 min, and the bacterial pellets were washed with PBS (three times) to obtain the final inoculum of 10^9^ CFU/0.2 mL. The sepsis model was based on a previous report [46], and the animals were inoculated intraperitoneally with 3 × 10^9^ CFU/0.2 mL.

In the abscess/necrosis model, the same procedure as with *P. aeruginosa* was followed, with the difference that 60 μL (1.5 × 10^9^ CFU) was inoculated into the right thigh of the animal, and the necrotic areas were measured up to 96 h.

In animal model assays, the incubation time of the inoculum with the plant extracts was approximately 20 min, at room temperature and without shaking. Subsequently, the viability of the inoculum was determined by the plate count method.

### 2.8. Statistical Analysis

The different parametric and non-parametric statistics used are indicated in detail in the figure captions. The experimental data were analyzed with SigmaPlot version 14.0 (Systat Software GmbH, Erkrath, Germany), and the multivariate statistical analysis was performed with R statistical software package (version 3.6.0; https://www.r-project.org/ accessed on 28 September 2021). The Kaplan–Meier curve was performed with GraphPad Prisma 6 (GraphPad, San Diego, CA, USA) and the data analysis in the SPSS 22.0 program (IBM Corporation, New York, NY, USA).

### 2.9. Ethical Declaration

All experiments with mice were carried out following the indications of the Research, Ethics and Biosafety Committees of the Hospital Infantil de México-Federico Gómez (HIM2018-002).

## 3. Results

### 3.1. Effect of Plant Extracts on the Pathogenicity of P. aeruginosa

The effect of 24 plant extracts (Table 1) at a sub-inhibitory concentration of 500 µg/mL (Appendix A) on the pathogenicity of *P. aeruginosa* was evaluated (Table 2, Appendix A). Subcutaneous inoculation of the PA14 WT strain induced the death of 42% (58% survival) of the animals, an abscess area of 255.8 mm^2^ at 24 h, and a necrotic area of 21.7 mm^2^ at 48 h. Additionally, at 96 h, establishment in the inoculation area was log_10_ 10.4 CFU/g and the systemic dispersion of log_10_ 3.2 CFU/g in the liver (Table 2). Interruption of the QS *(*Δ*lasR*/Δ*rhlR*) and T3SS *(*Δ*pscC*) systems increased the survival of the mice to 100% and reduced the pathogenicity of the bacteria, with the effect being more evident with the Δ*pscC* strain, which was not able to establish and generate damage (Table 2, Figure 1).

In the groups treated with the plant species, seven extracts (29%) increased the survival of the infected animals to 100%, highlighting *D. americana* and *H. sabdariffa* that also significantly reduced damage, establishment, and spread to the liver (Table 2, Figure 1 and Appendix A). *J. procumbens* significantly reduced damage but did not prevent establishment and spread to the liver (Table 2), while *P. coccineus* and *T. lucida* only increased survival and prevented dispersal to the liver (Table 2). In the case of *L. mexicana* and *P. peltilimba*, although they increased the survival of the animals, they did not reduce the establishment of the bacteria or the damage, and with *P. peltilimba*, a larger necrotic area of 44.8 mm^2^ was recorded (Table 2). Finally, six extracts allowed 66% survival, but only *G. viscosum*, *B. parviflora*, and *P. guajava* prevented the systemic dispersal of the bacteria (Table 2).

Interestingly, 11 extracts (45%) increased animal mortality relative to the group treated with PA14 WT. In seven treatments, a 33% survival was recorded, where *X. sagittifolium* increased the necrotic area by 40.3 mm^2^ (α < 0.05) (Table 2), while in the groups treated with *R. crispus*, *I. dumosa*, *A. glandulosum*-L/F, and *A. ludoviciana*, survival was 0% (Table 2, Appendix A).

#### 3.1.1. Effect of Plant Extracts on Virulence Factors Regulated by QS in *P. aeruginosa*

In the Δ*lasR*/Δ*rhlR* strain, pyocyanin production was reduced by 90%, while in the PA14 WT group treated with the positive control C-30 furanone, reduction was 50% (Figure 2A). In the case of the seven plant extracts that maintained 100% survival of the animals (Table 2), six (*P. peltilimba*, *P. coccineus*, *L. mexicana*, *J. procumbens*, *D. americana*, and *T. lucida*) were correlated with a decrease of up to 54% in pyocyanin production (α ≤ 0.05 *) (Figure 2A). *H. sabdariffa* stimulated pigment production by 14% (α ≤ 0.05 *) (Figure 2A). Interestingly, *R. crispus*, *I. dumosa*, *A. glandulosum*-L/F, and *A. ludoviciana*, which induced death in 100% of the animals, inhibited pyocyanin production by 17 to 48% (α ≤ 0.05 *) (Figure 2A). As expected, swarming was reduced by 92.6% in the Δ*lasR*/Δ*rhlR* strain and by 91.3% in the PA14 WT group treated with C-30 furanone (Figure 2B), while 91% of the extracts at 500 µg/mL strongly reduced swarming and only *G. glutinosum* and *S. scabrida* stimulated it from 10 to 35% at 125 and 250 µg/mL (α ≤ 0.05) (Figure 2B and Appendix A).

To determine the correlation between the variables involved in the effect of the extracts on pathogenicity, a principal component analysis (PCA) was carried out, and two groupings of different classes were identified. In the first place, the extracts with anti-pathogenic activity were related to increased survival and reduced necrosis, swarming, dissemination to the liver, and establishment of the bacteria in the inoculation area (Figure 3). Interestingly, as shown in principal component 2, the inhibition of pyocyanin does not reduce pathogenicity since it is the only variable that is not correlated (Figure 3). On the other hand, abscess formation correlated negatively with survival, which indicates it is a variable related to the increase in mortality and is placed where the second grouping of eight extracts is distributed (Figure 3). Finally, we identified five extracts whose behavior is not explained by the analyzed variables (Figure 3).

#### 3.1.2. Effect of Plant Extracts on T3SS of *P. aeruginosa*

We analyzed the effect of seven plant extracts that reduced pathogenicity in the animal model (as well as *R. crispus* that stimulated it) on ExoU secretion (Figure 4a).

The Δ*pscC* strain did not secrete ExoU, while the commercial inhibitor MBX-1641 (25 µM) used as a positive control completely inhibited it in the PA14WT strain (Figure 4a). Furthermore, in Δ*lasR*/Δ*rhlR*, ExoU secretion remains active (Appendix A). Additionally, five of the seven extracts (71%) at 100 µg/mL reduced ExoU secretion (Figure 4a) compared to *J. procumbens*, *P. peltilimba*, and *R. crispus* (Figure 4a, Table 2). The extracts of *D. americana* and *H. sabdariffa* that showed the best anti-pathogenic activity (Table 2, Figure 1) were the most efficient in reducing ExoU secretion in a dose–response manner (Figure 4b,c).

#### 3.1.3. Total Phenolic and Terpenoid Content

We analyzed the total content in seven plant extracts with the best anti-pathogenic activity (Table 1). In all the extracts, high flavonoid content was identified in the range of 64.35–85.00 mg quercetin equivalent/g of extract and terpenoids with 269.56–550.23 mg ursolic acid equivalent/g of extract (Appendix A). The concentration of terpenoids in *D. americana* is similar to that of *T. lucida*, *P. coccineus*, and *L. mexicana*, although its flavonoid content, like that of *H. sabdariffa*, is lower than that of *P. peltilimba* (*p* < 0.05) (Appendix A). Finally, for the case of the flavonoid HPLC profile of *D. americana*, it was possible to identify two peaks of greater intensity with UV spectra of the flavone type (λ_max_ 269/335 nm and 270/337 nm) (Appendix A).

### 3.2. Role of QS and T3SS in the Pathogenicity of C. violaceum

The abscess/necrosis model (*n* = 3 mice) was implemented with *C. violaceum* to investigate its potential in the preclinical evaluation of anti-virulence substances. Additionally, to analyze the participation of T3SS in pathogenicity, the mutant strain Δ*escU* was constructed. Likewise, 12 plant were selected, and their anti-pathogenic capacity was evaluated.

The intramuscular inoculation of 10^9^ CFU of the 31532 WT strain allowed establishment of the bacteria in the inoculation area, and generated abscesses at 24 h and necrosis at 48 h but did not cause death of the animals. Similarly, plant extracts at 500 µg/mL did not affect strain viability (Appendix A) nor did they reduce pathogenicity of the bacteria (Appendix A). For the case of the mutant strains, no significant differences were recorded in formation of abscesses and necrotic tissue (Appendix A).

We determined the effect of mutations on the production of violacein and some virulence factors. CviI synthase interruption did not alter swarming but reduced proteolytic activity (35%), biofilm formation (77.9%), and violacein production (95%), as well as hemolytic and chitinolytic activity (Table 3). In contrast, in Δ*escU* the production of these phenotypes was not significantly modified (Table 3, Appendix A).

We subsequently evaluated the effect of the strains on survival in a sepsis model in mice [46]. In the Kaplan–Meier curve, it was observed that the 31532 WT induced death of 50% of the animals 72 h after inoculation, and the trend was maintained up to 92 h, when the assay ended (Figure 5). This effect was counteracted when T3SS was blocked since 100% of the group inoculated with Δ*escU* survived (*p* = 0.001) (Figure 6). However, when CviI synthase was blocked (Δ*cviI*), there was a 25% increase in mortality relative to 31532 WT (*p* ≥ 0.05) and 75% with Δ*escU* (*p* < 0.0001) (Figure 5). In addition, analysis showed that median survival was higher with 31532 WT (72 h) than with Δ*cviI* (20 h).

A similar effect was recorded in the abscess/necrosis model using groups of five animals. The 31532 WT strain induced formation of 32.5 mm^2^ necrotic tissue and establishment of the bacteria in the inoculation area with log_10_ 5.2 CFU/g. In contrast, Δ*escU* induced a smaller abscess area with 102.9 mm^2^ (α ≤ 0.05) and necrosis with 3.12 mm^2^ (α ≤ 0.05), as well as lower establishment in the inoculation area with log_10_ 1.8 CFU/g (α ≤ 0.05) (Table 4 and Figure 5). However, Δ*cviI* induced a larger necrotic area of 40.6 mm^2^ (α ≤ 0.05) and establishment of log_10_ 6.2 CFU/g (α ≤ 0.05) (Table 4 and Figure 5). It should be noted that there was no systemic dissemination or death of the animals in any of the groups.

## 4. Discussion

The anti-virulence properties (mainly anti-QS) of substances of plant origin constitute a relatively new and rapidly developing field of research, which has positioned plant extracts as one of the top natural products with this activity [6,7]. In *P. aeruginosa*, pyocyanin and swarming are phenotypes commonly evaluated in this class of studies [17,18,19,27]. The first is a redox-active pigment that induces pro-inflammatory and oxidative effects that damage host cells [59], while swarming is a type of collective motility on semisolid surfaces that occurs as an adaptive response to microenvironmental changes [60].

With the abscess/necrosis model, we were able to determine the anti-pathogenic activity at the preclinical level of 23 edible and medicinal plant species (Appendix A) on *P. aeruginosa*, but of the two variables evaluated in vitro, only swarming inhibition was correlated with the variables in vivo (Figure 3). The *H. sabdariffa* extract stands out: it did not inhibit pyocyanin production but reduced pathogenicity of the bacteria in the animal model (similar to Δ*lasR*/Δ*rhlR*) and secretion of ExoU (Figure 5). Additionally, a group of four extracts was identified (*R. crispus*, *I. dumosa*, *A. glandulosum*-L/F, and *A. ludoviciana*) that reduced swarming and pyocyanin production (Figure 2) but increased animal mortality 100% (Table 2). It should be noted that the increase in mortality was not due to the possible toxicity of the extracts since their administration without bacteria did not cause death of the animals (Appendix A). These results indicate substantial differences between the anti-virulence activity of plant extracts recorded in vitro and their anti-pathogenic effect in the murine model. They also suggest that other elements or mechanisms of the host are involved in the regulation of virulence and are not present in in vitro assays. In this regard, some published studies have shown that QS systems can be modulated by host factors such as the microbiota, neurotransmitters, and stress hormones [61]. It was reported that norepinephrine interferes in QS regulation, favoring enteropathogenic *Escherichia coli* infection [62], while dynorphin A participates in activating the PQS system by potentiating *P. aeruginosa* virulence [63].

Additionally, environmental factors such as pH, temperature, and nutrient availability can influence QS modulation and virulence [64]. Two-component systems (TCSs) are one of the primary mechanisms used by bacteria to detect multiple environmental signals and generate an optimal response [65]. In *P. aeruginosa*, more than 50% of its TCSs are involved in virulence control or virulence-related behaviors [26].

As mentioned, ExoU is the most toxic effector secreted by the T3SS and an important anti-virulence target [66]. ExoU is a phospholipase that causes rapid cell lysis and necrosis in epithelial cells, but its activation requires host cofactors such as ubiquitin or ubiquitinated proteins [67]. This phospholipase generates an enriched environment of saturated fatty acids in the sputum of patients with cystic fibrosis [68]. However, it was reported that myristic acid reduces the secretion of ExoU in vitro (and other virulence factors regulated by QS) and in the abscess/necrosis model stimulates pathogenicity. Thus, it is suggested that the in vivo function of this saturated fatty acid is to act as an environmental signal molecule for the activation of virulence [69].

Most of the screenings of substances of plant origin have focused on demonstrating the inhibition of QS as the primary anti-virulence mechanism. However, in this research, we found that five of the seven extracts with the highest anti-pathogenic activity also reduce ExoU secretion (Figure 4). Thus, it is feasible that inhibition of multiple targets by different components of the extracts is what determines their anti-pathogenic capacity in the murine model. In this regard, it was reported that various derivatives of phenolic compounds control ExoS expression via the interference of a TCS called GacS-GacA [25]. Additionally, in a global analysis based on omics, it was found that citral (essential oil) acts through several mechanisms to reduce virulence (QS-regulated factors, type II and type IV secretion systems) of *A. baumannii* [28].

Thus, previous evaluation in the abscess/necrosis model helped to better identify plant extracts with anti-virulence activity against *P. aeruginosa*. We propose that it can be used in plant screening as a preliminary approach to in vitro studies. With this strategy, we identified two plant species with anti-pathogenic properties: *D. americana* and *H. sabdariffa*. In *D. americana* there are no studies on its bactericidal or anti-virulence properties. However, extracts from leaves of *D. robinioides* and *D. carthagenensis* are reported to be effective against enterobacteria and *Giardia lamblia* [70,71] due to the presence of diphysidione and vitexin [72]. In the case of vitexin, sub-inhibitory concentrations have been reported to reduce biofilm formation, swarming, and pyocyanin in *P. aeruginosa*. Furthermore, the molecular coupling analysis showed a strong binding affinity of this molecule with the LuxR, LasA, and LasI proteins [73]. In this study, we identified the presence of two major flavones in the extract of *D. americana*, so the anti-virulence activity of the extract may be related to the presence of vitexin (Appendix A). In *H. sabdariffa*, the antimicrobial properties of its calyces have been widely documented [74]. However, our research group recently identified hibiscus acid as one of the compounds responsible for anti-virulence activity since it reduced pathogenicity in the abscess/necrosis model [19].

On the other hand, *C. violaceum* is another widely used biosensor in identifying the anti-QS activity of plant extracts and phytochemicals at the in vitro level [75,76,77]. The abscess/necrosis model was implemented with this bacterium. Although the infection was achieved in small groups of mice (*n* = 3), no significant differences in the reduction of pathogenicity were observed between Δ*cviI* (Cv026) mutant strains and the plant extracts (Appendix A). In this regard, it has been shown that QS mutant strains of *P. aeruginosa* and *Staphylococcus aureus* have notably lower virulence and dispersion and cause lower mortality than the wild strain in various murine models [3,9,10,11]. Additionally, similar results have been obtained with the administration of phytochemicals that inhibit QS [15,19,49,50,51,73]. However, in the case of *C. violaceum*, the participation of the CviI/CviR system in pathogenicity in murine models had not yet been analyzed.

Infection with *C. violaceum* in animals is considered rare, and in humans, only in a few cases, the bacterium establishes and causes fatal septicemia [41]. Additionally, CviI/CviR has been reported to regulate various virulence factors [42,43,44]. The CviI enzyme synthesizes the autoinducer C6-acyl-homoserine lactone, and *cviR* encodes a cytoplasmic transcription factor called CviR. When the autoinducer concentration increases in the medium, it forms a protein–ligand complex with CviR in the cytoplasm and activates the transcription of the *vioABCDE* operon to produce violacein [42]. The CV026 *(*Δ*cviI*) strain is a widely used biosensor since it does not synthesize C6-acyl-homoserine lactone but expresses the *vioABCD* reporter gene if added exogenously to the medium because CviR remains active [78].

Additionally, *C. violaceum* has genes associated with two T3SSs located on different islands of pathogenicity called *Cpi-1/-1a* and *Cpi-2*, but only the first is necessary to generate liver damage and induce death in a mouse model of sepsis [46]. In this manner, in the construction of the Δ*escU* mutant strain, the *escU* gene of *Cpi-1/-1a* was deleted. This codes for a component of the export apparatus for the secretion of effector CipB, which induces the formation of pores in animal cells [46]. When analyzing the effect of these mutations, we corroborated that in Δ*cviI*, various virulence factors are attenuated or not produced, while in Δ*escU*, they remain active (Table 3).

In the sepsis model, only Δ*escU* did not cause death of the animals (Figure 5), which corroborates that T3SS is a virulence determinant in this model [46]. However, contrary to expectations, Δ*cviI* increased mortality in the sepsis model relative to the 31532 WT strain (Figure 5). It is important to note that the growth rate and the degradation pattern of sugars were the same in all strains (Appendix A).

Intraperitoneal inoculation of the bacteria might have been limiting CviI/CviR involvement. We consider that the abscess/necrosis model (subcutaneous inoculation) was more like the way *C. violaceum* infects through skin wounds [41]. However, we recorded a similar effect of Δ*cviI*, which was better established and generated more significant necrosis than 31532 WT (Table 4, Figure 6).

Of the few in vivo studies with *C. violaceum*, it was reported that the *C. elegans* model inoculated with Δ*cviI* delayed nematode death by an average of 12 days, while exogenous addition of C6-acyl-homoserine lactone restored pathogenicity, and animals died within 4 h [79]. Nevertheless, one explanation for reducing Δ*cviI* pathogenicity in the *C. elegans* model may be related to the nematicide activity of violacein [80].

On the other hand, the importance of the different virulence factors may lie in the ecological niche in which the bacteria develop. The main niche of *C. violaceum* is in swampy soils, where the production of violacein and other phenotypes regulated by QS allow it to compete for space and resources with other microorganisms [45], while according to our results in animal models, they are not determinants in establishment and damage.

However, T3SS is a virulence determinant with the ability to cause harm and death. In this regard, it has been shown that in other bacterial species, negative regulation can occur between the QS and T3SS systems [81]. In this study, we could not assess whether the Δ*cviI* strain had increased effector production caused by possible down-regulation with T3SS. Additionally, in bacterial isolates from lesions, contaminating colonies were commonly present, and it is thus possible that exogenous autoinducers activated the QS of Δ*cviI*. However, although our results indicate that *cviI* is not a pathogenicity determinant in murine models, studying this phenomenon in greater depth is necessary.

In conclusion, the abscess/necrosis model makes it possible to evaluate the anti-virulence capacity of plant extracts at the preclinical level on *P. aeruginosa*. Contrary to expectations, the results suggest that the elimination of *cviI* in the QS system may favor the pathogenicity of *C. violaceum* in murine models. Finally, studies are needed to understand the host factors that drive global virulence to develop more effective anti-virulence therapies.

## Figures and Tables

**Figure 1 microorganisms-09-02424-f001:**
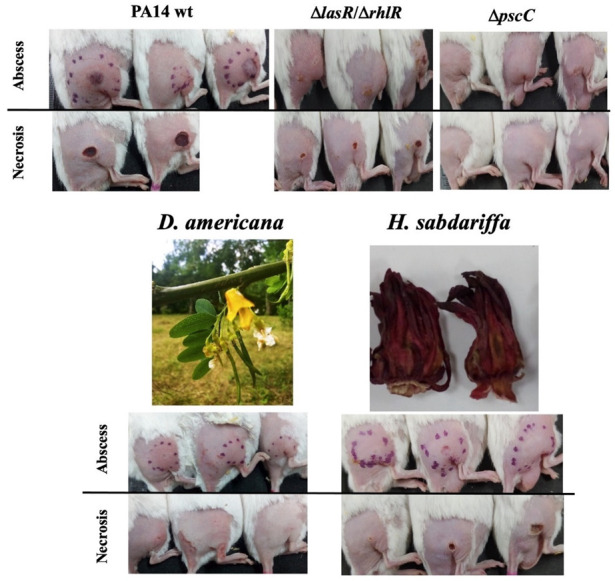
Representative images of the effect of mutations in QS *(*Δ*lasR*/Δ*rhlR*) and T3SS *(*Δ*pscC*) of *P. aeruginosa* and the anti-pathogenic activity of dichloromethane extracts of *D. americana* (pods) and *H. sabdariffa* (calyxes) at a sub-inhibitory concentration of 500 µg/mL.

**Figure 2 microorganisms-09-02424-f002:**
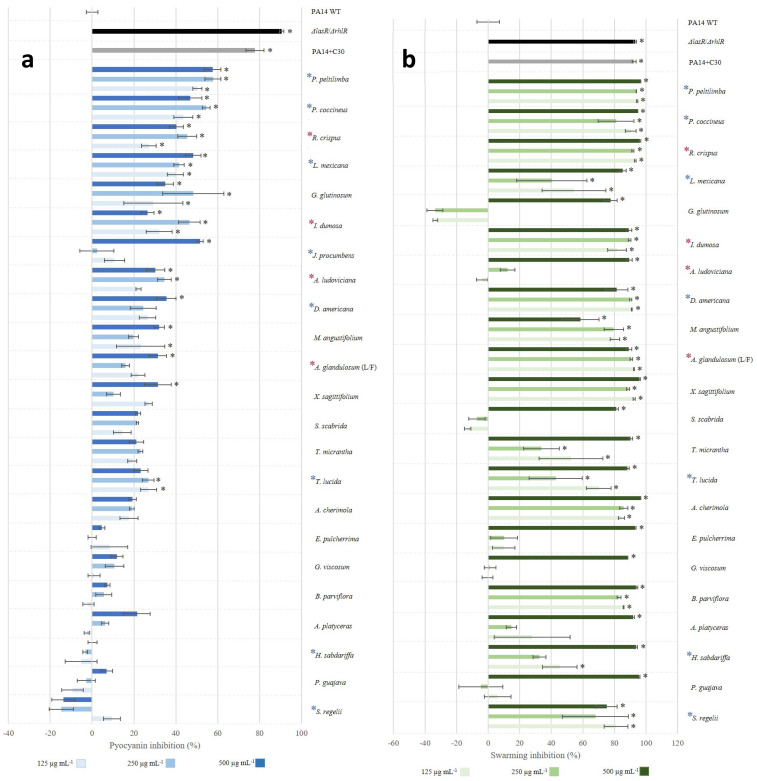
Anti-virulence activity of dichloromethane extracts on *P. aeruginosa*. (**a**) Pyocyanin inhibition and (**b**) swarming. The data represent the mean and the standard deviations of two repetitions with *n* = 5. Significant difference from the wild type (Kruskal–Wallis, *p* ≤ 0.05 *, and Student–Newman–Keuls test, α ≤ 0.05 *). C-30, furanone C30 (50 µM). Blue asterisks indicate the main extracts that reduced pathogenicity and red asterisks indicate extracts that stimulated it in the animal model.

**Figure 3 microorganisms-09-02424-f003:**
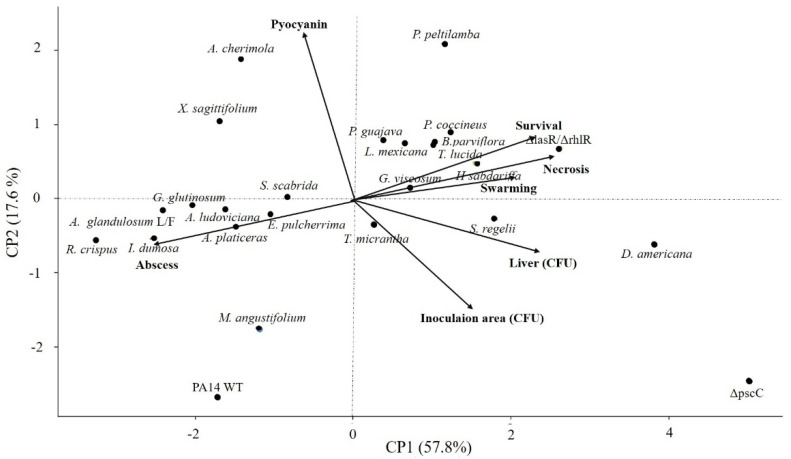
Principal component analysis of anti-virulence activity (in vitro) and anti-pathogenic capacity (abscess/necrosis model) of plant extracts in *P. aeruginosa*. The groups of extracts that reduce pathogenicity and those that stimulate it are shown. Mutant strains are also represented, in which the variance dispersion of the Δ*lasR*/Δ*rhlR* is more like the group of extracts with anti-pathogenic activity. The values were transformed into percentages, and the variables were managed under the inhibition premise, except for survival.

**Figure 4 microorganisms-09-02424-f004:**
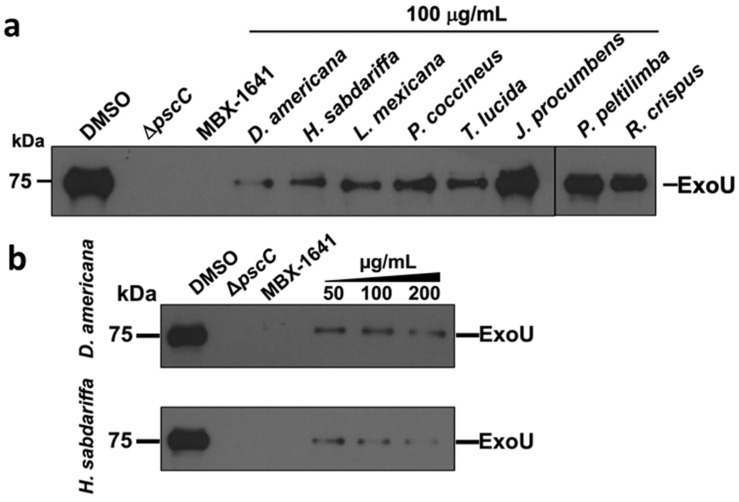
Effect of plant extracts on ExoU protein secretion (Western blot) in *P. aeruginosa*. MBX1641 25 µM (ChemBridge) is a T3SS inhibitor that was used as a positive control. (**a**) Representative image of two independent tests of seven extracts that showed the best anti-pathogenic activity in vivo and that of *R. crispus* which stimulated death of the animals. (**b**) Dose–response effect in reducing effector secretion by *D. americana* and *H. sabdariffa* extracts. Band intensity is quantified in (**c**); bands represent the average of three independent tests (Kruskal–Wallis, *p* ≤ 0.05, and Student–Newman–Keuls test, α ≤ 0.05 *). The maximum concentration used was 200 µg/mL because higher concentrations presented solubility problems in the assay.

**Figure 5 microorganisms-09-02424-f005:**
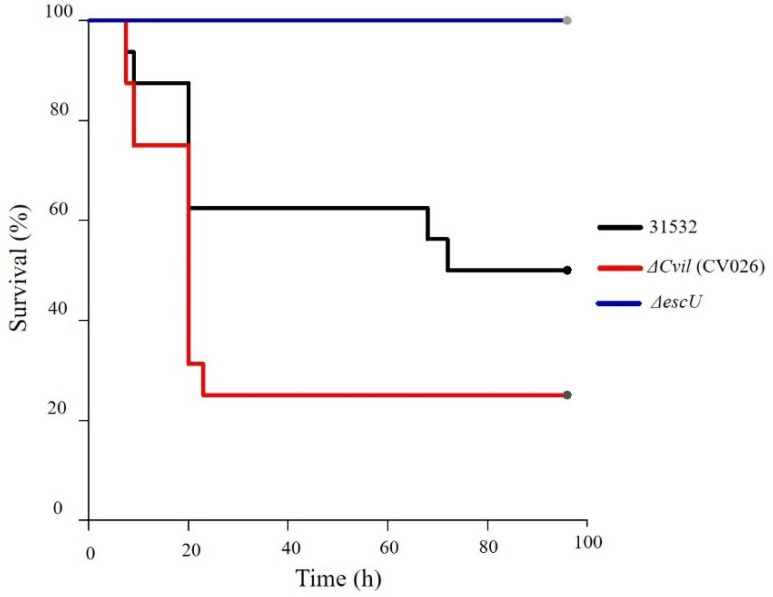
Survival of mice infected with *C. violaceum* in the sepsis model. Animals were injected intraperitoneally with 3 × 10^9^ CFU/0.2 mL of each of the strains. The data are representative of two independent experiments with groups of eight mice. The Kaplan–Meier curve was performed with the GraphPad Prisma 6 program (95% confidence interval) and the data analysis in the SPSS 22.0 program.

**Figure 6 microorganisms-09-02424-f006:**
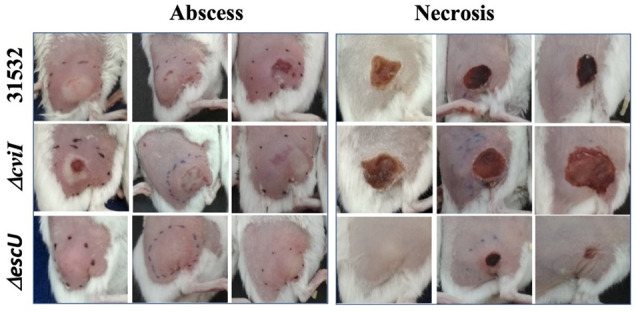
Representative images of abscesses (24 h) and necrosis (96 h) caused by strains of *C. violaceum* in mice. The animals were injected in the subcutaneous zone with 1.5 × 10^9^ CFU/0.06 mL of the different strains.

**Table 1 microorganisms-09-02424-t001:** Plant species used in this study.

Species Name (Family)	Local Name	Folk Usage	Plant Part Extracted	Voucher #
*Allium glandulosum* Link & Otto (Amaryllidacae)	Cebolla de monte	Edible	Bulbs and Leaves/flowers	155,009
*Annona cherimola* Mill. (Annonaceae)	Chirimoya	Edible, insecticide, and medicinal	Leaves	155,006
*Argemone platyceras* Link & Otto (Papaveraceae)	Chicalote, amapola silvestre, cardo santo	Medicinal and ornamental	Whole plant	155,007
*Artemisa ludoviciana* (Asteraceae)	Hierba maestra	Medicinal	Whole plant	155,015
*Buddleja parviflora* H.B. & K (Scrophulariaceae)	Tepozán	Medicinal	Whole plant	155,014
*Diphysa americana* (Mill.) M. Sousa (Fabaceae)	Quebrache	Edible	Pods	154,998
*Euphorbia pulcherrima* Willd. ex Klotzsch (Euphorbiacea)	Flor de nochebuena	Medicinal and ornamental	Leaves	155,008
*Gnaphalium viscosum* Kunth (Asteraceae)	Gordolobo	Medicinal	Whole plant	155,013
*Gymnosperma glutinosum* (Spreng.) Less. (Asteraceae)	Tatalencho	Medicinal	Whole plant	155,018
*Hibiscus sabdariffa* L. (Malvaceae)	Jamaica	Edible	Calyxes	*
*Ipomoea dumosa* (Benth.) L.O. Williams (Convolvulaceae)	Soyo	Edible	Leaves	154,993
*Jaltomata procumbens* (Cav.) J.L. Gentry (Solanaceae)	Jaltomate	Edible	Fruits	155,004
*Loeselia mexicana* (Lam.) Brand (Polemoniaceae)	Espinosilla	Medicinal	Whole plant	155,012
*Metastelma angustifolium Turcz* (Apocynaceae)	Not available	Medicinal	Whole plant	155,017
*Peperomia peltilimba* C.DC. ex Trel. aff. *Peperomia aggravescens* Trel. (Piperaceae)	Tequelite	Edible	Edible	154,994
*Phaseolus coccineus* L. (Fabaceae)	Frijol ayocote	Edible	Flowers	154,992
*Psidium guajava* L. (Myrtaceae)	Guayaba	Edible	Whole plant	Not available
*Rumex crispus* L. (Polygonaceae)	Quelite agrio	Edible	Leaves	154,996
*Saurauia scabrida* Hemsl. (Actinidiaceae)	Acalama	Edible	Fruits	155,002
*Smilax regelii* Killip & C.V. Morton (Smilacaceae)	Cocolmeca	Edible	Stems	154,999
*Tagetes lucida* Cav. (Asteraceae)	Pericón	Medicinal and edible	Whole plant	155,016
*Tagetes micrantha* Cav. (Asteraceae)	Anicillo	Medicinal and edible	Whole plant	155,011
*Xanthosoma sagittifolium* (L.) Schott (Araceae)	Quelite de lampazo	Edible	Leaves	155,005

* Previous study [19]. **#** number.

**Table 2 microorganisms-09-02424-t002:** Effect of plant extracts at sub-inhibitory concentration on the pathogenicity of *P. aeruginosa*.

Treatments	Survival (%)	Abscess Area (mm^2^, Mean ± S.E.)	Necrotic Area (mm^2^, Mean ± S.E.)	Bacterial Load	(log_10_ CFU/g)
				Inoculation Area	Liver
PA14 WT	58	255.8 ± 26.9	21.7 ± 3.7	10.4 ± 0.08	3.2 ± 0.05
Δ*lasR*/Δ*rhlR*	100	56.4 ± 44.9 *	6.2 ± 2.4 *	6.6 ± 2.2 *	0.7 ± 0.7 *
Δ*pscC*	100	0 *	0 *	0 *	0 *
PA14 WT + plant extract (500 μg/mL)					
*D. americana*	100	66.0 ± 48.3 *	1.6 ± 0.2 *	0 *	0 *
*H. sabdariffa*	100	130.1 ± 53.0	7.5 ± 3.6 *	8.5 ± 0.9 *	0 *
*J. procumbens*	100	64.1 ± 40.0 *	5.9 ± 3.7 *	9.9 ± 0.4	2.9 ± 0.08
*P. coccineus*	100	131.0 ± 24.8	15.8 ± 5.4	9.1 ± 0.3	0 *
*T. lucida*	100	146.0 ± 13.5	19.3 ± 1.1	9.1 ± 0.9	0 *
*L. mexicana*	100	124.0 ± 14.4	21.4 ± 8.3	9.9 ± 0.3	2.0 ± 1.0
*P. peltilimba*	100	55.8 ± 28.6 *	41.8 ± 28.0 *	6.9 ± 3.5	3.1 ± 0.7
*S. regelii*	66	53.5 ± 30.5 *	13.0 ± 6.5	5.7 ± 2.9 *	2.9 ± 1.0
*G. viscosum*	66	158.3 ± 80.4	16.0 ± 13.1	7.8 ± 0.5 *	0 *
*B. parviflora*	66	107.2 ± 32.0 *	14.3 ± 2.5	10.1 ± 0.2	0 *
*P. guajava*	66	151.4 ± 71.2	19.9 ± 5.5	9.8 ± 0.08	0 *
*A. glandulosum*-B	66	212.5 ± 68.8	9.9 ± 6.9	10.2 ± 0.02	2.9 ± 0.7
*A. cherimola*	66	168.4 ± 1.6	51.4 ± 29.4 *	10.2 ± 0.1	3.2 ± 0.2
*T. micrantha*	33	172.5 ± 38.4	10.2 ^#^	7.8 ^#^	0 ^#^
*M. angustifolium*	33	218.4 ± 45.4	10.8 ^#^	10.2 ^#^	2.9 ^#^
*A. platyceras*	33	208.0 ± 36.8	12.8 ^#^	10.3 ^#^	3.6 ^#^
*E. pulcherrima*	33	192.7 ± 96.4	15.4 ± 15.4	8.9 ^#^	3.4 ^#^
*S. scabrida*	33	149.2 ± 41.6	24.5 ^#^	9.9 ^#^	3.3 ^#^
*G. glutinosum*	33	227.5 ± 8.7	33.9 ^#^	10.4 ^#^	3.2 ^#^
*X. sagittifolium*	33	190.7 ± 22.4	40.3 ± 14.5 *	10.2 ^#^	2.8 ^#^
*R. crispus*	0	310.9 ± 28.6	0 ^#^	-	-
*I. dumosa*	0	252.1 ± 11.8	21.1 ^#^	-	-
*A. glandulosum*-L/F	0	237.5 ± 31.5	-	-	-
*A. ludoviciana*	0	176.9 ± 21.1	0 ^#^	-	-

The animals were inoculated subcutaneously with 10^7^ CFU of the different strains. By counting standard plates, the viability of PA14 WT was determined after incubating them for 20 min at room temperature with the plant extracts. The area of the abscess was quantified 24 h after inoculation and the formation of necrosis at 48 h, which corresponds to the maximum time of its formation. The animals were sacrificed at 96 h, and the survival percentage and the CFU in the tissues were determined. The experiment was carried out once with groups of three animals per treatment, except the control groups with PA14 WT, which were two independent tests with three and four animals (Appendix A). * Significant difference from wild type (*p* ≤ 0.05; Kruskal–Wallis and α ≤ 0.05; Student–Newman–Keuls test). ^#^ Data are from a single animal so no statistical analysis could be performed. B: bulb; and L/F: leaves and flowers. - no data.

**Table 3 microorganisms-09-02424-t003:** Effect on the production of violacein and virulence factors in QS and T3SS mutant strains of *C. violaceum*.

Phenotype	31532 WT	Δ*cviI* (CVO26)	Δ*escU*
Violacein	100 ± 22.5 ^a^	4.7 ± 1.1 ^b^	77.7 ± 30.6 ^a^
Chitinolytic activity	100 ± 2.4 ^a^	0 ± 0 ^b^	101.6 ± 3.4 ^a^
Biofilm formation	100 ± 38 ^a^	22.1 ± 3.4 ^b^	135 ± 53.7 ^a^
Proteolytic activity	100 ± 1 ^a^	65 ±1 ^b^	99.4 ± 1.2 ^a^
Hemolytic activity	100 ± 2.6 ^a^	0 ± 0 ^b^	99.5 ± 2.0 ^a^
Swarming	100 ± 2.5 ^a^	99.6 ± 4.1 ^a^	100 ± 1.9 ^a^

± S.E., one-way ANOVA and Tukey comparative test. Means without a common letter significantly differ, *p* < 0.05.

**Table 4 microorganisms-09-02424-t004:** Role of QS and T3SS of *C. violaceum* in pathogenicity in abscess/necrosis model.

Strain	Abscess Area (mm^2^, Mean ± S.E.)	Necrotic Area (mm^2^, Mean ± S.E.)	Bacterial Load in the Inoculation Area (log_10_ CFU/g)
31532 WT	145.03 ± 5.06 ^a^	32.54 ± 4.03 ^a^	5.3 ± 0.70 ^a^
Δ*cviI* (CV026)	144.61 ± 8.7 ^a^	40.69 ± 5.07 ^a^	6.26 ± 0.47 ^a^
Δ*escU*	103 ± 6.2 ^b^	3.12 ± 1.38 ^b^	1.83 ± 0.69 ^b^

The abscess area was determined at 24 h and the necrotic area at 96 h. On the fourth day post-infection, the injured tissue and the liver were obtained to determine the number of CFU/g. The experiment was carried out three times with five mice per group. One-way ANOVA and Tukey comparative test. Means without a common letter significantly differ, *p* < 0.05.

## Data Availability

Not applicable.

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
