# Peer review of "Anti-Virulence Properties of Plant Species: Correlation between In Vitro Activity and Efficacy in a Murine Model of Bacterial Infection"

_microorganisms, 2021, doi:10.3390/microorganisms9122424_

Round 1

Reviewer 1 Report

Please, see the attached report.

Author Response

The authors thank you for your valuable comments on our papers. 

Reviewer 2 Report

It is interesting to note that plant extracts exhibit anti-virulence activity in a mouse model of abscess and necrosis induced by Pseudomonas aeruginosa. However, I think there is a problem with the way this paper is organized. In the experimental results using Chromobacterium violaceum, no activity of the plant extract was detected, and the relationship between the anti-virulence activity exhibited by the plant extracts and the inhibition of bacterial quorum sensing is unclear. In particular, the experimental results presented in Tables 3 and 4 and Figures 5 and 6 have no relevance to the effects of the plant extracts, and in my opinion, are an out-of-place presentation of the data, considering the main purpose of this paper. The logic of this paper to link the anti-virulence activity of the plant extracts to the inhibition of quorum sensing is not supported by the experimental facts and gives a misleading impression. I request the authors to revise the paper to take this into account.

Author Response

The authors thank you for your valuable comments on our paper. 

Reviewer 3 Report

Quorum sensing (QS) is a microbial system that uses small signaling molecules to mediate cell-to-cell communication. These molecules keep symbiotic homeostasis among microorganisms and can control various microbial virulence factors. Thus their proper communication can support host health, but their dysbalance can contribute to various illnesses. Therefore the control of QS is one of the promising ways to combat infectious diseases in the situation of multidrug resistance of microbial pathogens and restriction of using classical antibiotics.

José Luis Díaz Núñez et al. submitted their manuscript entitled „Anti-virulence properties of plant species: correlation between in vitro activity and efficacy in a murine model of bacterial infection“ to Microorganisms. The manuscript covers the study of the effect of plant extracts on bacterial QS and its anti-virulence effect.

The manuscript is interesting, deals with the actual topic, and is thoroughly elaborated. Thus, I have a few notices only.

L131: Please complete 60° to 60°C.

P138-139: Ideally, wild-type (WT) P. aeruginosa and WT-derived mutants should be used. What was the reason that WT was not used?

L147: The used abbreviation WT should be introduced on line 140.
L174: 21,1000 - please, correct the number.
L206-209: Rearing conditions influence experimental results. It should be specified size of groups and (commercial) diet at least.

L220: Here should be written 60 ul of inoculum containing 10^7 CFU of P. aeruginosa.
L229: Did you evaluate that sepsis was really induced? Did you use any sepsis markers?
L272: Three animals per treatment is a deficient number. Hardly apply any statistical comparison.
L287-289: Any explanation should be in discussion but not in the description of results. Please, remove this explanatory text.

Figures 2a and 2b should have larger text. Their increase made the description blurred.

L332-333: The first sentence of the paragraph should be part of the discussion but not the description of the results.

L353-354: The first sentence of the paragraph should be part of the discussion but not the results.
L356: Please, modify to 64.35-85.00 mg ...

L364-365: The first sentence of the paragraph should be in the discussion but not in the results.

L375-377: The first sentence of the paragraph should be in the discussion.

Producers are usually introduced as (Name of company, City of the headquarter or branch, Country), e.g. (Bio-Rad, Hercules, USA) when they are first mentioned. It is possible to abbreviate affiliation in repeated use, e.g. (Bio-Rad). It would be very suitable to keep common customs.

Author Response

The auhors thank you for your valuable comments on our paper. 

Round 2

Reviewer 2 Report

As far as the author's opinion is concerned, the evaluation of this paper is going to be left to the reader. I think there is nothing wrong with the experimental data and the argument. I agree that this paper should be accepted.

Author Response

The authors are grateful for your comments and suggestions to our paper that have enriched and improved it.